# Mycophagous rove beetles highlight diverse mushrooms in the Cretaceous

Chenyang Cai[1,2], Richard A.B. Leschen[3], David S. Hibbett[4], Fangyuan Xia[5] & Diying Huang[2]

Agaricomycetes, or mushrooms, are familiar, conspicuous and morphologically diverse Fungi. Most Agaricomycete fruiting bodies are ephemeral, and their fossil record is limited. Here we report diverse gilled mushrooms (Agaricales) and mycophagous rove beetles (Staphylinidae) from mid-Cretaceous Burmese amber, the latter belonging to Oxyporinae, modern members of which exhibit an obligate association with soft-textured mushrooms. The discovery of four mushroom forms, most with a complete intact cap containing distinct gills and a stalk, suggests evolutionary stasis of body form for ~99 Myr and highlights the palaeodiversity of Agaricomycetes. The mouthparts of early oxyporines, including enlarged mandibles and greatly enlarged apical labial palpomeres with dense specialized sensory organs, match those of modern taxa and suggest that they had a mushroom feeding biology. Diverse and morphologically specialized oxyporines from the Early Cretaceous suggests the existence of diverse Agaricomycetes and a specialized trophic interaction and ecological community structure by this early date.

[1] Key Laboratory of Economic Stratigraphy and Palaeogeography, Nanjing Institute of Geology and Palaeontology, Chinese Academy of Sciences, Nanjing 210008, China. [2] State Key Laboratory of Palaeobiology and Stratigraphy, Nanjing Institute of Geology and Palaeontology, Chinese Academy of Sciences, Nanjing 210008, China. [3] Landcare Research, New Zealand Arthropod Collection, Private Bag 92170, Auckland, New Zealand. [4] Department of Biology , Clark University, Worcester, Massachusetts 01610, USA. [5] Lingpoge Amber Museum, Shanghai 201108, China. Correspondence and requests for materials should be addressed to D.H. (email: dyhuang@nigpas.ac.cn).

Agaricomycetes is the most conspicuous and morphologically diverse group of Fungi[1]. Most agaricomycete fruiting bodies are ephemeral[2], and so their fossils are extremely sparse[2–6]. Evidence indicating the origin and early diversification of Agaricomycetes is very limited. A Jurassic fossil that had been interpreted as a bracket fungus[7] was shown to be the outer bark of a conifer[8]. To date, five definitive species of agarics (gilled mushrooms) have been known exclusively from amber. Among them, two different forms are from the Mesozoic, including the earliest mushrooms, *Palaeoagaracites antiquus* from mid-Cretaceous Burmese amber[3] (~99 Myr old), and the slightly younger *Archaeomarasmius leggetti* from New Jersey amber[2,4] (~90 Myr old). The remaining three species, *Aureofungus yaniguaensis*[5], *Coprinites dominicana*[6] and *Protomycena electra*[4], are known from early Miocene Dominican amber, some 20 Myr old. All known fossil agarics are very small in size. Here we report four new forms of modern-looking gilled mushrooms (Agaricales) and diverse mycophagous rove beetles (Coleoptera, Staphylinidae) from mid-Cretaceous Burmese amber, the latter belonging to Oxyporinae, modern members that exhibit an obligate association with mature soft-textured mushrooms[9–11]. The specialized mouthpart morphology of these beetles sheds light on the early evolution of insect–fungal associations. More importantly, diverse and morphologically specialized oxyporines from the Early Cretaceous[12,13] suggest a probable occurrence of diverse large-sized Agaricomycetes by that period.

## Results

**Studied material.** The material includes fossil mushrooms and beetles: five mushrooms of four distinctive forms (Taxa A–D) in Burmese amber (~99 Myr old) from Hukawng Valley, northern Myanmar, and five species and four genera of oxyporine beetles. The beetles consist of two new *Oxyporus* species (Taxa 1 and 2) and a new genus (Taxon 3) from Burmese amber, and two monotypic genera (*Protoxyporus* and *Cretoxyporus*) from the Lower Cretaceous Yixian Formation (~125 Myr old) of northeastern China. These fossils are extremely rare among the 111,000 Burmese amber inclusions and in our collections of the Nanjing Institute of Geology and Palaeontology, Chinese Academy of Sciences.

**Diverse gilled mushrooms from Burmese amber.** The mid-Cretaceous fossil mushrooms (Fig. 1; Supplementary Fig. 1) are clearly Agaricomycetes, a derived group of fungi that plays significant ecological roles as decomposers, pathogens, and symbionts in terrestrial ecosystems and that includes most edible mushrooms. Three of the four mushrooms (Fig. 1a,c,e; Supplementary Figs 2a,c and 3a,b) are nearly complete, with an intact cap (pileus), gills (lamellae) and stalk. All are minute, with caps ranging from 2.6 to 3.9 mm in diameter. The caps (Fig. 1a,c–e; Supplementary Figs 2a,c and 3a–c) range from strongly to slightly plano-convex and are mostly radially sulcate. Lamellae (Fig. 1b; Supplementary Figs 2b,d and 3e) are mostly sub-distant and comparatively close in one form. Macromorphological features of these fossils resemble extant mushrooms. In particular, two of four forms (Taxon A and B; Fig. 1a,c; Supplementary Fig. 2a,c) are similar to the extant genera *Marasmius*, *Marasmiellus* or *Crinipellis*, and the fossil *Archaeomarasmius* from late-Cretaceous New Jersey amber, suggesting that they belong to the family Marasmiaceae (Agaricales). The other two forms are difficult to place in extant families due to the lack of micromorphological features and inadequate preservation. Taxon C (Fig. 1d; Supplementary Fig. 3c) has a slightly convex pileus (Fig. 1d), close lamellae (Supplementary Fig. 3e) and a sub-marginal stalk (Supplementary Fig. 3d), a combination of features that is not easy to compare with modern agarics. Taxon D (Fig. 1e; Supplementary Fig. 3a,b) has a plicate-pectinate cap margin, which is similar to that of the Miocene *Coprinites* from Dominican amber. However, the former differs from *Coprinites* by the short, stout and sub-marginal stalk (Supplementary Fig. 3b). The discovery of four mushroom forms from Burmese amber, together with the known *Palaeoagaracites antiquus* from the same deposit, highlights the palaeodiversity of Agaricomycetes in the mid-Cretaceous. Like their modern counterparts in Marasmiaceae, these mushrooms (Taxa A and B) were probably decayers of leaf litter and wood in ancient ecosystems. Detailed descriptions of the fossil mushrooms are given in Supplementary Note 1.

**Mycophagy and mycophagous oxyporine rove beetles.** Mycophagy, or fungus-feeding, is widespread in Coleoptera[14–16] and the occurrence of this feeding habit in older clades of many lineages[17–19] suggests that it preceded phytophagy (feeding on

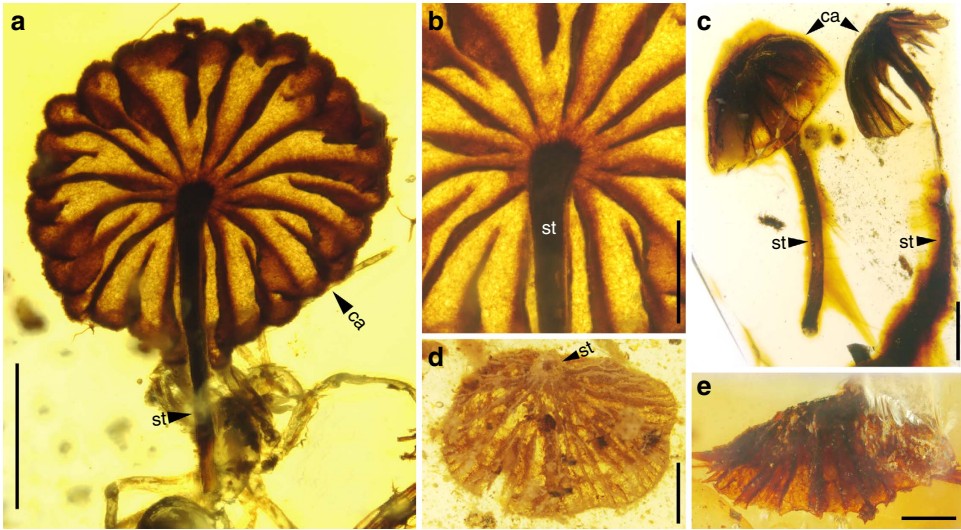

**Figure 1 | Diverse mushrooms in mid-Cretaceous amber from northern Myanmar.** (**a**) General habitus of Taxon A, FXBA10101, ventral view. (**b**) Enlargement of **a**, showing details of lamellae and top portion of stalk. (**c**) Lateral view of two individuals of Taxon B, NIGP164521 (left) and NIGP164522 (right). (**d**) Ventral view of Taxon C, NIGP164523, showing sub-marginally inserted stalk. (**e**) Lateral view of Taxon D, NIGP164524. Abbreviations: ca, cap; st, stalk. Scale bars, 1 mm (**a**,**c**,**d**); 500 μm (others).

plant tissues). Specialized feeding on mushrooms (including Agaricales, Boletales and Polyporales) occurs in a few beetle lineages and involves species that feed on spores and conidia or those that feed on the hymenium or hyphal tissue, each correlated with specialized mouthparts[17,20]. Recent fossil-based findings shed light on this feeding behaviour, and we report diverse specialized obligately mycophagous rove beetles (Fig. 2a–i; Supplementary Figs 4–9) that set an early date for mushroom specialization and evidence for the existence of diverse Agaricomycete fruiting bodies in the Early Cretaceous.

These brown to black beetles, with a body length ranging from 4.5 to 7.9 mm (Fig. 2a–c; Supplementary Figs 4, 6a,b and 7a), are conspicuous among all staphylinids discovered in Burmese amber. The beetles are definitely oxyporine rove beetles (Staphylinidae: Oxyporinae) as evidenced by the characteristic mouthparts (enlarged mandibles, greatly enlarged apical labial palpomeres; Fig. 3a–e; Supplementary Figs 5a,b,e–i, 6d–f, 7b–f

and 8a,b,e), and widely separated mesocoxae present in extant forms (see Supplementary Note 2 for detailed description). These oxyporines include three distinctive species, with two (Taxa 1 and 2) assignable to the extant *Oxyporus* and the third belonging to a new genus (Taxon 3; see Supplementary Note 2).

The most remarkable structure of these beetles is the long, enlarged and anteriorly extended mandibles with the incisor edge well developed. The left mandible (Fig. 3b; Supplementary Fig. 8c) has a notch-like structure on the outer ventral margin to receive the right mandible when at rest just as that found in extant *Oxyporus* species (Supplementary Figs 10a and 11a,b)[10,11,21]. The right mandible (Supplementary Fig. 8c), unlike modern *Oxyporus*, has a jagged incisor edge with dense small, sharp anteriorly directed teeth and a distinct process forming a deep notch-like structure on the ventral margin to receive the left mandible and its ventral notch-like structure. Similar-looking small teeth are found on the mandibles in other soft-tissue

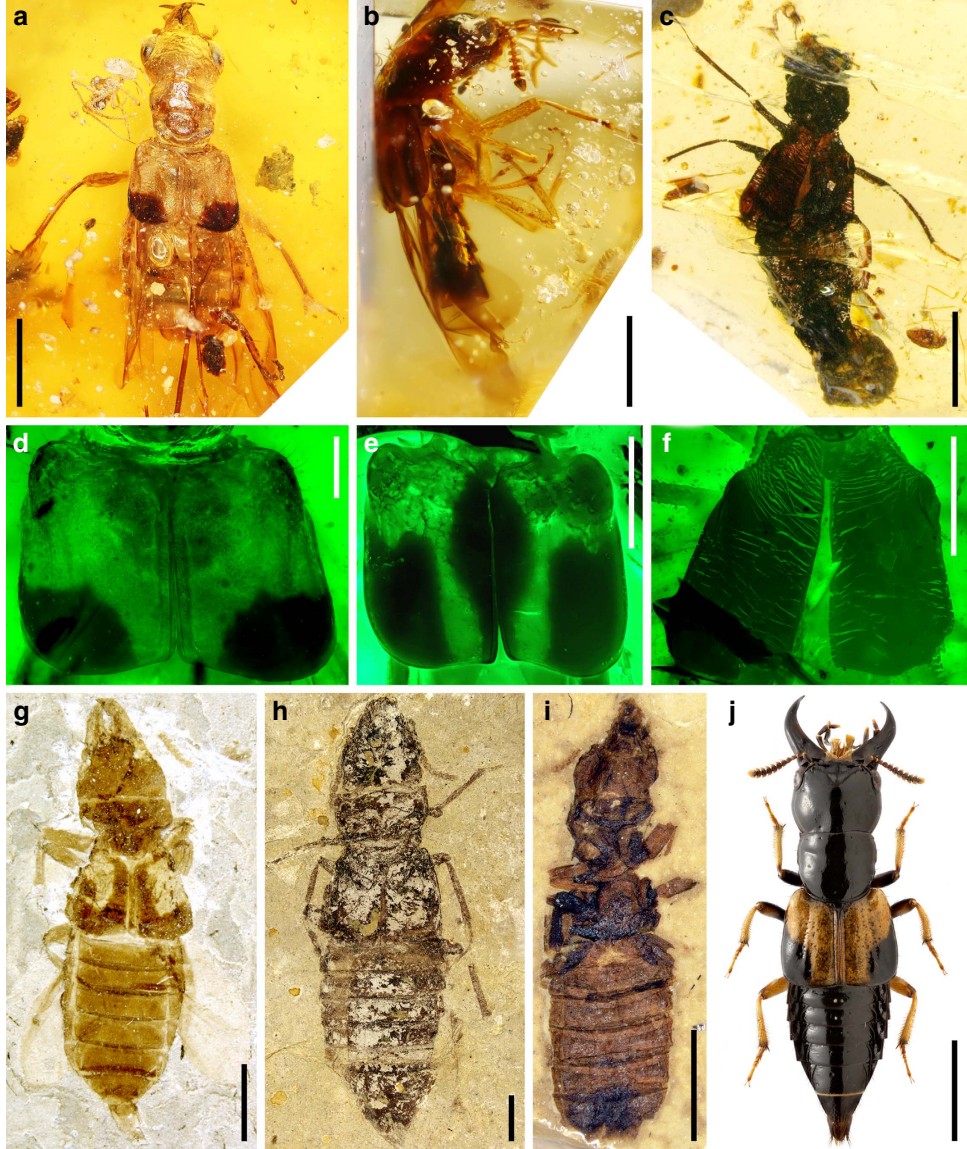

**Figure 2 | Diverse mycophagous oxyporine rove beetles.** (**a**–**c**) Beetles from mid-Cretaceous Burmese amber, (**d**–**f**) under fluorescence, (**g**–**i**) from the Early Cretaceous Yixian Formation of northeastern China. (**a**) Dorsal view of Taxon 1, NIGP164526. (**b**) Lateral view of Taxon 2, NIGP164528. (**c**) Dorsal view of Taxon 3, NIGP160556. (**d**) Enlargement of elytra from **a**. (**e**) Enlargement of elytra from **b**. (**f**) Enlargement of elytra from **c**. (**g**) *Oxyporus yixianus*; image courtesy of Yanli Yue. (**h**) *Protoxyporus grandis*. (**i**) *Cretoxyporus extraneus*. (**j**) *O. maxillosus*; image courtesy of Maxim Smirnov. Scale bars, 2 mm (**a,c** and **g**–**j**); 1 mm (**b**); 500 μm (**d**–**f**).

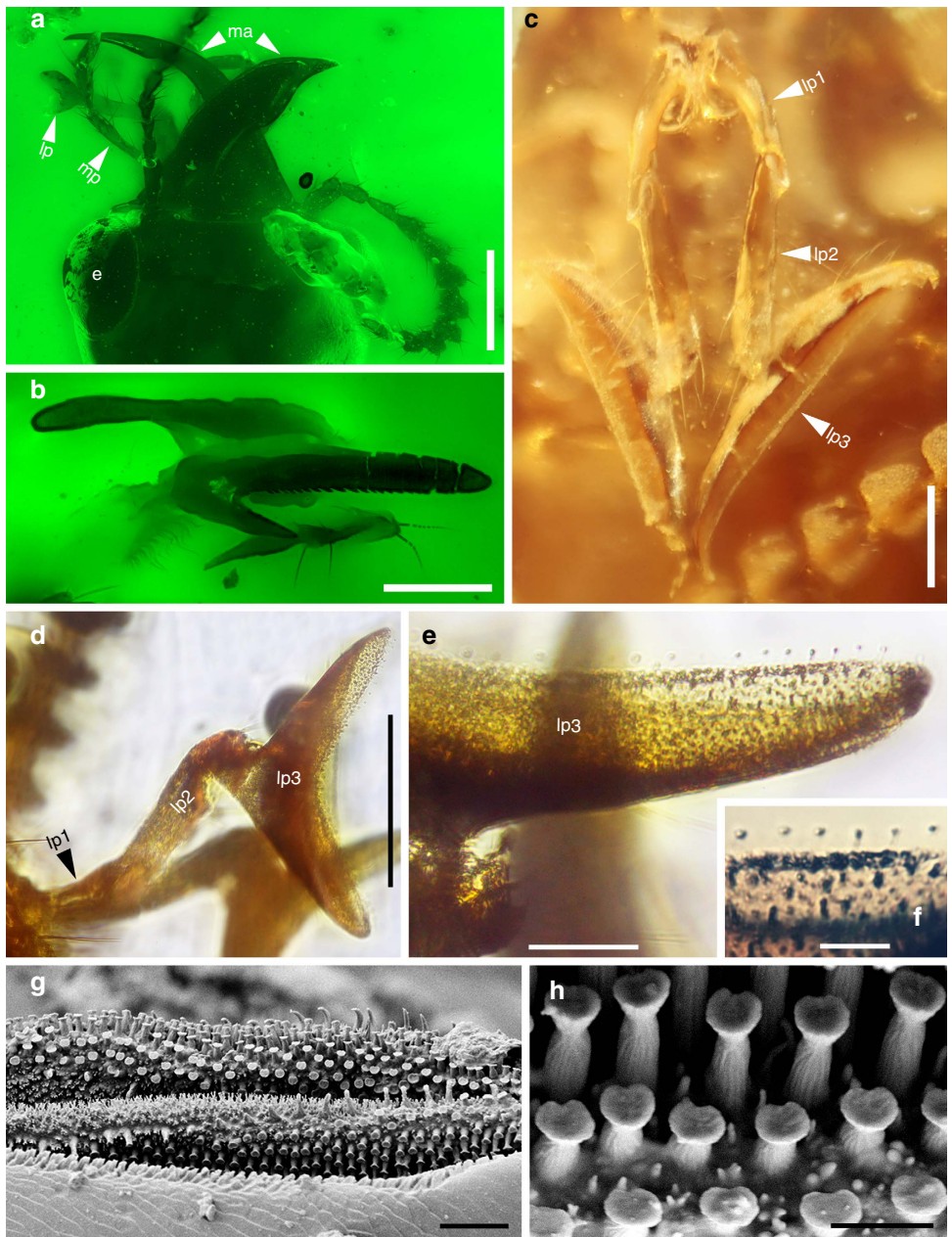

**Figure 3 | Details of mouthparts of extinct and extant mycophagous oxyporine rove beetles.** (**a**,**b**) Images under fluorescence, (**c**) under reflected light, (**d**–**f**) under transmitted light and (**g**,**h**) under SEM. (**a**) Mouthparts of Taxon 1, NIGP164527. (**b**) Mandibles of Taxon 3, NIGP160556. (**c**) Labial palpi of Taxon 1, NIGP164526. (**d**) Labial palpus of Taxon 2, NIGP164528. (**e**) Enlargement of **d** showing dense peg-like sensory organs. (**f**) Enlargement of **e**, showing details of sensory organs. (**g**) Apex of labial palpus of extant *Oxyporus* sp., showing both peg-like and villiform sensory organs. (**h**) Details of peg-like sensory organs. Abbreviations: e, eye; lp, labial palpomere; ma, mandible; mp, maxillary palpomere. Scale bars, 500 μm (**a**); 200 μm (**b**–**d**); 50 μm (**e**); 20 μm (**f** and **g**); 5 μm (**h**).

specialists, including Scaphidiinae mushroom feeders[20] and the modifications probably represent one of the principal morphological adaptations of Cretaceous oxyporines to mushroom feeding as in extant *Oxyporus* species[10]. Like modern *Oxyporus* (Supplementary Fig. 11a,c,e–h), the mandibles bear a basal pseudomola (Supplementary Fig. 8b), or prostheca, in the form of a brush, a structure assumed as an adaptation to pre-oral digestion in *Oxyporus* adults[22]. The ventral basal area of the mandibles possesses a brush-like structure on the posterior region that serves to increase the surface area for masticating the fungal slices and possibly mixing the material with digestive enzymes produced from the gut. In addition, the closed mandibles probably serve to form a container for the

bolus as suggested in its present-day counterparts[23], while the notch-like structures on both mandibles are slightly out-of-line and probably serve as a compression device during mandibular apposition to further macerate fungal tissue.

Another impressive feature of these early beetles is the highly modified labial palpi. The labial palpi (Fig. 3c,d; Supplementary Figs 7f and 8e) are three-segmented, with the apical segment laterally compressed and crescent-shaped (Fig. 3d; Supplementary Fig. 7g), a characteristic feature of modern Oxyporinae (Supplementary Figs 9d, 10a–d, 11d and 12a,b)[10,11] but also found in some staphylinine staphylinids[10]. The first labial palpomere is distinctly shorter than the second, which bears a shallow anterolateral notch at the apex for receiving the base of the

apical palpomere (Fig. 3c,d; Supplementary Fig. 7f). The last labial palpomere is widened and possesses an apical enlarged surface covered with dense, fine peg-like structures (Fig. 3e,f), sometimes appearing as darkened spots from certain angle (Supplementary Fig. 7h) and similar to structures found in extant oxyporines (Fig. 3g,h; Supplementary Fig. 12c–f), although at least three other types of sensory organs (sensillae) are detected from the latter (Supplementary Fig. 12g,h). The sensory areas may aid in the recognition of its host fungi or evaluate quality of the fungal host and are features that also occur on the palpi of obligate mushroom-feeding Erotylidae beetles[10]. Oxyporinae are thought to be members of a predatory group of staphylinines exhibiting different methods of prey handling, including the bizarre stick–capture method for prey–capture by stenines[24,25]. However, the long-held view that the ancestral Oxyporinae had shifted from predation to mushroom feeding is compromised by different placements in recent phylogenetic studies[26,27] (Supplementary Note 3). The exact phylogenetic placement of Oxyporinae remains controversial, complicating the exact nature of the origin of mushroom feeding in the group. Oxyporinae are placed in the predatory Staphylinine group of subfamilies, in a basal position relative to Megalopsidiinae[26], but recently the Staphylinine group is recovered as polyphyletic, and Oxyporinae as a sister to Leptotyphlinae[27]. The peculiar mouthpart structure of fossil oxyporines suggests that the beetles may have been feeding on soft tissues like fleshy mushrooms as do modern oxyporines[10,18], though some of these features are not exclusive to fungus feeding lineages[23] and that host-shifts among widely different food types may occur as long as the texture of the substrates is similar[20].

## Discussion

Most extant Agaricomycetes have ephemeral fruiting bodies, although the group also includes taxa with tough, persistent sporocarps, like those of wood-decaying polypores[28]. The fossil record of Agaricomycetes is limited, with only five definitive species of agarics (gilled mushrooms) known previously and four new forms reported here (Supplementary Table 1). All known fossil agarics are small in size, with the pileus ranging from 2.2 to 5.0 mm in diameter (Supplementary Table 1). Their small size and life habits including growing on certain resin-producing plants probably contribute significantly to their amber fossilization. Many extant Marasmiaceae have tough stipes and pilei that can shrivel on drying but then revive on rewetting, which may also promote preservation as fossils. Assuming that the Burmese oxyporines were mushroom specialists with similar habits as the modern species, such as subsocial care on large mushroom fruiting bodies[10], it is unlikely that they fed on the contemporaneous mushrooms from the same deposit described herein, primarily due to the small-sized fruiting body with a pileus < 4 mm in diameter. Both larvae and adults of modern *Oxyporus* species construct tunnels in the mushroom cap upon which they feed[9,22,29] and adult females construct brood chambers[30]. The body size of fossil oxyporines were comparatively large (4.5–7.9 mm long), and to construct tunnels and build brood chambers to accommodate up to eight or more eggs[23,30] in fossil mushrooms much smaller than the beetle body length seems unlikely. Therefore, it is likely that during the mid-Cretaceous the Agaricomycetes were more diverse than previously documented and included large mushrooms, related to, but distinct from, the fossil mushrooms reported here.

In addition to multiple oxyporine forms in Burmese amber, oxyporine beetles are known from the older Yixian Formation (*ca.* 125 Myr) of northeastern China (Supplementary Table 2), including *Oxyporus yixianus* (Fig. 2g), *Protoxyporus grandis* (Fig. 2h) and *Cretoxyporus extraneus* (Fig. 2i; Supplementary Fig. 9a–c). All these beetles bear the characteristic oxyporine body shape and prominent mandibles, though the expanded labial

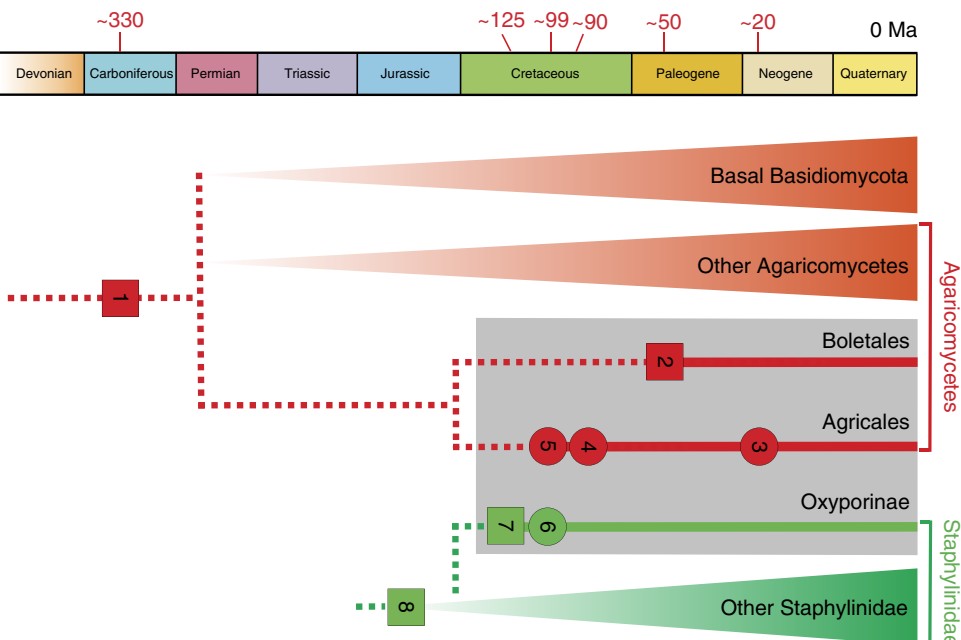

**Figure 4 | Associations of higher Agaricomycetes and specialized oxyporine Staphylinidae.** Framework for Basidiomycota (red part) based on Taylor and Berbee[31], Floudas *et al.*[36] and Hibbett *et al.*[37] 1: Oldest basidiomycete clamp connections from late Visean (Mississippian, ~330 Myr) of France[38]; 2: oldest Boletales (Ectomycorrhizae) from middle Eocene (~50 Myr) Princeton chert of British Columbia[39]; 3: modern-appearing mushrooms from Miocene (~20 Myr) Dominican amber[3–5]; 4: mushrooms from the Late Cretaceous (~90 Myr) New Jersey amber[1,3]; 5: diverse mushrooms from mid-Cretaceous (~99 Myr) Burmese amber; 6: diverse obligately mycophagous Oxyporinae from mid-Cretaceous Burmese amber; 7: diverse Oxyporinae from the Early Cretaceous Yixian Formation (~125 Myr) of China[1]; 8: oldest known Staphylinidae from the Middle Jurassic (Aalenian – Bathonian) of Kubekovo, Russia[40]. Squares, compression fossils. Circles, amber inclusions.

palpus is present only in *C. extraneus* (Supplementary Fig. 9c). Many mushroom beetles are aposematic, having warning colours that indicate toxicity to potential predators. This is true for many modern *Oxyporus* (Fig. 2j), which are often gaudy and bi- or tricoloured[11]. Although one of the Burmese oxyporines has black elytra (Fig. 2f), the other two species (Fig. 2d,e; Supplementary Figs 5d and 6b,c) and the older fossils (*Protoxyporus* and *O. yixianus*) have distinct bi-coloured elytra, suggesting mushroom-related biology in the Cretaceous. Among Early Cretaceous oxyporines, *P. grandis*, with a body length of 19.7 mm, represents the largest and most conspicuous of the oxyporines, a giant among the entire subfamily when compared with the modern oxyporines that range from 5.5 to 13.0 mm in length[11]. Modern large-sized *Oxyporus* species (>8 mm long), including *O. major* (8.2–12.7 mm long) and *O. rufipennis* (8.5–13.0 mm long), appear to have preferences for large fleshy mushrooms such as *Pleurotus ostreatus* (oyster mushroom, cap 5–25 cm across), *Polyporus squamosus* (8–30 cm across), *Armillaria gallica* (*ca.* 10 cm across), and some *Boletus* species[10,30], though there are a few records for some species that may be associated with large aggregations of moderately sized fungi[22]. Therefore, it is possible that the large-bodied Cretaceous *P. grandis* was associated with large-sized fruiting bodies, such as those produced by extant Agaricales, Boletales or Polyporales, that would accommodate larval growth, especially if these species were subsocial. In addition to having larger body sizes in the Early Cretaceous (8.1–19.7 mm long) the extinct species differ in several significant features, including cephalic and mesocoxal structures, which may indicate that they were adapted to different types of mushrooms present in the Early Cretaceous (Supplementary Fig. 13). Molecular clock dating studies have yielded highly inconsistent age estimates for the Fungi, with the Basidiomycota inferred to have originated from 450 Myr ago to over 1.1 Gyr ago[31–35]. A recent genome-based molecular clock analysis with fossil-based calibrations estimated the mean age of the Agaricomycetes as *ca.* 290 Myr ago[36]. Thus, it is probable that associations between specialized oxyporine rove beetles and Agaricomycetes were well established in the Early Cretaceous (Fig. 4), consistent with the hypothesis that higher fungi, including the main groups of mushrooms, had already diversified by the Early Cretaceous[31,36].

## Methods

**Specimen preparation and imaging.** All specimens (except FXBA10101) are housed at Nanjing Institute of Geology and Palaeontology, Chinese Academy of Sciences; FXBA10101 is housed in the Lingpoge Amber Museum in Shanghai. The amber has been polished with sand papers with different grain sizes and diatomite mud. Photomicrographs were taken using the Zeiss Discovery V20 microscope system, and those with green background (Figs 2d–f and 3a,b; Supplementary Figs 2d,e, 5a,c,d, 6a,c and 8) were using fluorescence as light source attached to a Zeiss Axio Imager 2 compound microscope. Finally, a compression fossil (NIGP153699) was examined with a LEO1530VP field emission scanning electron microscope.

**Data availability.** All data generated during this study are included in this published article (and its Supplementary Information files).

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

## Acknowledgements

We are grateful to H.-L. Shi for providing extant oxyporine beetles. Financial support was provided by the Strategic Priority Research Program (B) of the Chinese Academy of Sciences (XDB18000000), the Ministry of Science and Technology (2016YFC0600406), the National Natural Science Foundation of China (41688103, 41602009 and 91514302), and the National Natural Science Foundation of Jiangsu Province (BK20161091). R.A.B. Leschen was supported in part by Core funding for Crown Research Institutes from the Ministry of Business, Innovation and Employment's Science and Innovation Group. Anne Austin reviewed an early draft of the manuscript.

## Author contributions

C.C., R.A.B.L., F.X. and D.H. participated in morphological studies; D.H. designed the program; C.C., D.S.H. and R.A.B.L. prepared the manuscript.

## Additional information

**Competing interests:** The authors declare no competing financial interests.

**Publisher's note**: 

