## [Peer Review File · Nature Communications]

Reviewers' comments:

Reviewer #1 (Remarks to the Author):

The paper reports interesting and important findings, both for fungi and rove beetles. It is well executed paper, clearly written and sufficiently illustrated. Co-evolutionary considerations are sound. The manuscript contains several minor inconsistencies which are flagged and commented in the attached PDF. Supplementary data well document the taxa. However, assessment of the existing controversies of the sister-group relationships of Oxyporinae is written too superficially. Since the origin of oxyporines is important for the high ambition goal of the paper, phylogenetic literature that so far contributed to understanding the Oxyporinae sister-group relationships and remaining controversial must be summed up and evaluated more rigorously.

Also the title of the paper is misleading. It assumes more broad review of beetles. It should be restricted to Staphylinidae.

The paper is worth to be published in peer review journal after addressing these comments by the authors.

Reviewer #2 (Remarks to the Author):

This article describes several new fossils of basidiomycetes and mycophagous beetles from Cretaceous amber and other deposits. The findings are both novel and important and they are discussed in relation to what is previously known about the evolution of these two groups.

While there most probably was no direct ecological link between these particular fungi and insects, discussing them together is in principal well justified and provides new multidisciplinary insight into this interesting and poorly known topic.

However, the curious reader is left to wonder whether also other groups of mushroom-associated insects are known from the same amber material? The authors mention that A) oxyporine beetles are extremely rare among the thousands of Burmese amber inclusions and that B) also other presumably mycophagous insects are known from the Cretaceous, but they do not directly state whether other mycophagous insects are known to occur Burmese amber. If such fossils were to exist, it might be a bit arbitrary to only link the description of the fungi to oxyporine beetles and not discuss them in a wider feeding guild context.

The manuscript is for most part well written and illustrated. However, I was left wondering about the 'symmetry' of evolutionary lineages in Figure 4 where the fungi are tentatively traced back to the Ediacaran, but the beetles seem to 'pop up from nowhere' in the Triassic/Jurassic.

Maybe also the beetles should be tentatively traced further back with a similar dashed line – as there is no real difference in the extent of the fossil record between the two lineages. Or would it be better to leave Paleopyrenomycites (Ascomycota) out of the figure (and maybe only mention it in the legend?) and start the timeline from the Devonian – and place the box of the clamp connection in a 'fuzzy region' at the base of the different Basidiomycete lineages or something along these lines? In any case, the average reader needs more background information to understand the figure: why are the groups 'Basal Basidiomycota' and 'Other Agaricomycetes' included and so forth?

I presume that the tentative divergence times of the lineages in Figure 4 are based on the (highly

relevant) study of Taylor & Berbee 2006 as it is the only dating study mentioned in the references? This should be made clear in the figure legend. Finally, why did the authors not want to cite more recent dating studies on fungi – many readers might find a (very short) summary of them helpful in this context.

Response to Referees:

To Reviewer #1:

The paper reports interesting and important findings, both for fungi and rove beetles. It is well executed paper, clearly written and sufficiently illustrated. Co-evolutionary considerations are sound.

A: Thank you for your constructive feedback on our paper. We have addressed each of your comments in the revised manuscript. A detailed point-by-point response is listed as follows (in blue).

The manuscript contains several minor inconsistencies which are flagged and commented in the attached PDF. Supplementary data will document the taxa.

A: Thanks. All minor inconsistencies have been corrected. See below for details.

However, assessment of the existing controversies of the sister-group relationships of Oxyporinae is written too superficially. Since the origin of oxyporines is important for the high ambition goal of the paper, phylogenetic literature that so far contributed to understanding the Oxyporinae sister-group relationships and remaining controversial must be summed up and evaluated more rigorously.

A: Thank you for bringing this point to our attention. We have provided discussion on this fact in the main text, and more detailed in the 'Supplementary Information'. Two important related references are cited (as refs. 26 and 27 in the main text; one based on morphology, and the other based on molecular data). We also provided a paragraph dealing with this issue in 'Supplementary Information', i.e., the last paragraph of the section '3. Evolution of mycophagy in Staphylinidae'. The following sentences are added to the main text:

[The exact phylogenetic placement of Oxyporinae remains controversial, complicating the exact nature of the origin of mushroom feeding in the group. Oxyporinae are placed in the predatory Staphylinine group of subfamilies, in a basal position relative to Megalopsidiinae²⁶, but recently the Staphylinine group is recovered as polyphyletic, and Oxyporinae as a sister to Leptotyphlinae²⁷.]

Also the title of the paper is misleading. It assumes more broad review of beetles. It should be restricted to Staphylinidae.

A: Yes, we now can understand the reviewer's concern. We have changed 'beetles' to 'rove beetles (Staphylinidae)' to make the title more specific. The new title is 'Mycophagous rove beetles highlight diverse mushrooms in the Cretaceous'.

The paper is worth to be published in peer review journal after addressing these comments by the authors.

Specific linguistic points raised by Reviewer #1 (from the annotated PDF):

1. Line 47: 'taxon' is singular. 'taxa' is plural. Use correctly.

A: 'taxon' is changed to 'taxa'.

2. Line 78: There is growing trend in the phylogenetic literature to avoid using expressions 'basal' or 'more basal' in the sense of time. Here you refer to the clades that are earlier in origin, or branched off earlier, or older, etc. So, use these expressions.

A: OK. 'basal' used here is changed to 'older'.

3. Line 123: This is implicit reference to the Staphylininae-group of subfamilies introduced by Newton a while ago. There is a growing evidence that this group is not monophyletic. And position of Oxyporinae is especially problematic. So, here use a more neutral term 'group' instead, since 'lineage' means 'monophyletic group'.

A: Yes, we agree that there is evidence suggesting that Staphylininae-group of subfamilies is not monophyletic. 'lineage' is now changed to 'group'.

4. Line 130: 'texture of substrates' would be correct way to write this here, I think.

A: OK, '...as long as the textures of the substrate are similar' is modified to '...as long as the texture of the substrates is similar'.

5. Line 141: insert reference to publications reporting that.

A: OK, ref. 10 is added to supporting this statement.

6. Line 170: bring consistency. Two lines below you write 'Early Cretaceous'.

A: Yes, 'early-Cretaceous' is changed to 'Early Cretaceous'.

7. Line 299: beetles

A: OK, 'beetle' is changed to 'beetles'.

To Reviewer #2:

This article describes several new fossils of basidiomycetes and mycophagous beetles from Cretaceous amber and other deposits. The findings are both novel and important and they are discussed in relation to what is previously known about the evolution of these two groups.

A: Thank you for your constructive suggestions. We have addressed each of your comments in the revised manuscript. A detailed point-by-point response is listed as follows (in blue).

While there most probably was no direct ecological link between these particular fungi and insects, discussing them together is in principal well justified and provides new multidisciplinary insight into this interesting and poorly known topic.

A: Thanks.

However, the curious reader is left to wonder whether also other groups of mushroom-associated insects are known from the same amber material? The authors mention that A) oxyporine beetles are extremely rare among the thousands of Burmese amber inclusions and that B) also other presumably mycophagous insects are known from the Cretaceous, but they do not directly state whether other mycophagous insects are known to occur Burmese amber. If such fossils were to exist, it might be a bit arbitrary to only link the description of the fungi to oxyporine beetles and not discuss them in a wider feeding guild context.

A: Thank you for bringing this point to our attention. No mushroom-associated insects have been known or published from the Burmese amber. Among all known fossil insects from Burmese amber, only oxyporine rove beetles are known to be obligate mushroom feeders. We also mentioned in the Supplementary Information that definitive mushroom feeding by adults and larvae occurs in Oxyporinae and Scaphidiinae and a definitive unpublished scaphidiine has recently been discovered from our Burmese amber collection. However, although Scaphidiines are strictly mycophagous, they are not obligate Agaricales (mushrooms) feeders, and some feed exclusively on the spores of myxomycetes (e.g., Leschen, R. A. B., & Löbl, I., 1995. Phylogeny of Scaphidiinae with redefinition of tribal and generic limits (Coleoptera: Staphylinidae). *Revue Suisse de Zoologie*, 102, 425-474).

In addition, as pointed out by Reviewer #1, the title of the paper has been changed to 'Mycophagous rove beetles highlight diverse mushrooms in the Cretaceous'. This title is more specific and would avoid confusions.

The manuscript is for most part well written and illustrated. However, I was left wondering about the

‘symmetry’ of evolutionary lineages in Figure 4 where the fungi are tentatively traced back to the Ediacaran, but the beetles seem to ‘pop up from nowhere’ in the Triassic/Jurassic. Maybe also the beetles should be tentatively traced further back with a similar dashed line – as there is no real difference in the extent of the fossil record between the two lineages.

A: We apologize for the confusion in the previous version of Fig. 4. The hypothesis that the fungi are tentatively traced back to the Ediacaran is based on fossil-calibrated molecular clock analyses by different studies (e.g., Taylor & Berbee, 2006; Berbee & Taylor, 2001; Floudas et al., 2012). The green part of Fig.4 focuses on the family Staphylinidae. Based on the fossil record and DNA-based divergence time estimates of Staphylinidae, Staphylinidae probably have originated about 193 million years, close to the Triassic/Jurassic boundary (McKenna et al., 2015). In order to make Fig.4 easier to understand, we have provided three important references (refs. 31, 36 and 37) in the figure caption and a short summary concerning phylogenetic relationships and recent divergence time estimates of Basidiomycota in the main text.

The Staphylinidae part (in green) has been also modified: the family can be tentatively traced back to the Triassic/Jurassic boundary, which is symbolized with a similar dashed line. In addition, the oldest known Staphylinidae (and relevant reference, ref. 40) is added near the base of the clade.

References:

- Taylor, J.W., Berbee, M.L., 2006. Dating divergences in the fungal tree of life: review and new analyses. *Mycologia* 98: 838–849.
- Berbee, M. L., & Taylor, J. W. (2001). Fungal molecular evolution: gene trees and geologic time. In *Systematics and evolution* (pp. 229–245). Springer Berlin Heidelberg.
- Floudas D, et al. 2012. The Paleozoic origin of enzymatic lignin decomposition reconstructed from 31 fungal genomes. *Science* 336: 1715–1719
- Mckenna, D. D., Wild, A. L., Kanda, K., et al., 2015. The beetle tree of life reveals that Coleoptera survived end-Permian mass extinction to diversify during the Cretaceous terrestrial revolution. *Systematic Entomology*, 40(4), 835–880.

Or would it be better to leave *Paleopyrenomycites* (Ascomycota) out of the figure (and maybe only mention it in the legend?) and start the timeline from the Devonian – and place the box of the clamp connection in a ‘fuzzy region’ at the base of the different Basidiomycete lineages or something along these lines? In any case, the average reader needs more background information to understand the figure: why are the groups ‘Basal Basidiomycota’ and ‘Other Agaricomycetes’ included and so forth?

A: Thank you very much for the nice idea. The Ascomycota fossil (*Paleopyrenomycites*) has been totally removed from Fig. 4, and the new figure focuses on the Basidiomycota fungi and the Oxyporinae beetles. The red box of the clamp connection (Box 1) is now in a ‘fuzzy region’ at the base of the Basidiomycete. As also mentioned above, we have provided three related references concerning the phylogenetic relationships and recent divergence time estimates of Basidiomycota both in the main text and the figure caption, which would make both ‘Basal Basidiomycota’ and ‘Other Agaricomycetes’ understandable for average readers.

I presume that the tentative divergence times of the lineages in Figure 4 are based on the (highly relevant) study of Taylor & Berbee 2006 as it is the only dating study mentioned in the references? This should be made clear in the figure legend.

A: Yes, it was our mistake not to give the original source. Three relevant references (refs. 31, 36 and 37), including Taylor & Berbee (2006), has been added to the figure legend.

Finally, why did the authors not want to cite more recent dating studies on fungi – many readers might find a (very short) summary of them helpful in this context.

A: Yes, thanks, it is a good idea. A short summary of molecular clock dating studies have been given

near the end of the paper. In addition to Taylor and Berbee (2006), five more relevant references (refs. 32–36, see below) are added to the Reference list.

References:

Douzery, E. J., Snell, E. A., Baptiste, E., Delsuc, F. & Philippe, H. The timing of eukaryotic evolution: does a relaxed molecular clock reconcile proteins and fossils? *Proc. Natl. Acad. Sci. USA* 101, 15386–15391 (2004).

Berbee, M. L. & Taylor, J. W. Dating the molecular clock in fungi—how close are we? *Fung. Biol. Rev.* 24, 1–16 (2010).

Gueidan, C., Ruibal, C., De Hoog, G. S. & Schneider, H. Rock-inhabiting fungi originated during periods of dry climate in the late Devonian and middle Triassic. *Fungal Biol.* 115, 987–996 (2011).

Lücking, R., Huhndorf, S., Pfister, D. H., Plata, E. R. & Lumbsch, H. T. Fungi evolved right on track. *Mycologia* 101, 810–822 (2009).

Floudas, D., et al. The Paleozoic origin of enzymatic lignin decomposition reconstructed from 31 fungal genomes. *Science* 336, 1715–1719 (2012).

REVIEWERS' COMMENTS:

Reviewer #1 (Remarks to the Author):

The authors addressed well all points raised in both reviews and significantly improved the manuscript. I fully support its publication in Nature Communications.

Alexey Solodovnikov (reviewer 1).

Response to the referees:

To Reviewer #1:

The authors addressed well all points raised in both reviews and significantly improved the manuscript. I fully support its publication in Nature Communications. Alexey Solodovnikov (reviewer 1).

A: Thanks again for your constructive comments and suggestions that greatly improved our manuscript.